# Fac-TDMPC: A Factored World Model for Robot Planning

**Yuan Zhang**                                                    *yzhang@cs.uni-freiburg.de*
*University of Freiburg, Germany*

**Jianhong Wang**                                              *jianhong.wang@bristol.ac.uk*
*University of Bristol, UK*

**Jinke He**                                                              *j.he-4@tudelft.nl*
*Delft University of Technology, Netherlands*

**Frans A. Oliehoek**                                              *f.a.oliehoek@tudelft.nl*
*Delft University of Technology, Netherlands*

**Joschka Boedecker**                                          *jboedeck@cs.uni-freiburg.de*
*University of Freiburg, Germany*

**Reviewed on OpenReview:** *https://openreview.net/forum?id=Smb0sdocmd*

## Abstract

Model-based reinforcement learning (MBRL) has shown strong sample efficiency in robotics by learning predictive world models and planning with them, but existing methods suffer from high planning latency due to the combination of centralized world models and model predictive control (MPC) as planners, thus limiting the real-time deployment in high-dimensional action spaces. We introduce **Fac-TDMPC**, a factored latent-space world model that decomposes transition, reward, and value functions on the latent space and learns the factorization via model distillation. The factored design enables decentralized planning across action dimensions. Empirically, Fac-TDMPC achieves substantial planning speedups while preserving the control performance across a suite of continuous-control robotic tasks; it also demonstrates improved robustness to action perturbations, interpretable joint-level latent structure, and enhanced multi-task data efficiency.

## 1 Introduction

Reinforcement learning (RL) (Sutton & Barto, 1998) is a powerful paradigm for solving sequential decision-making problems, achieving remarkable success across diverse domains, such as mastering the game of Go (Silver et al., 2017) and controlling quadruped robots (Kostrikov et al., 2023). Model-based reinforcement learning (MBRL) (Moerland et al., 2023) is a prominent branch of RL that offers superior sample efficiency and improved interpretability compared to the model-free approaches (Wang et al., 2019; Ye et al., 2021). TDMPC (Hansen et al., 2022) and its successor TDMPC2 (Hansen et al., 2023) represent state-of-the-art MBRL methods in robotics. These frameworks learn a world model for representing robotic systems in latent spaces and perform trajectory optimization using model predictive control (MPC) in the latent space, enabling long-horizon planning and constraint handling.

Despite their successes, TDMPC-based frameworks typically employ a centralized world model with densely connected neural networks as dynamics, reward and value functions, resulting in an MPC optimization space of $\mathcal{O}(|\mathcal{A}|^H)$, where $\mathcal{A}$ denotes the action space and $H$ is the planning horizon. For the high-dimensional

action space, this leads to substantial computational overhead during planning, often referred to as *the curse of dimensionality* (Bertsekas, 2019), which restricts their scalability to more complex robotic scenarios (e.g. humanoid benchmark (Sferrazza et al., 2024)) and real-time deployment. Notably, in many real-world scenarios, planning over the entire action space is unnecessary, since the effects of different actions on the environment are often largely uncorrelated. For instance, a common approach to humanoid whole-body control is to decompose the upper and lower body, assigning balance and manipulation tasks to the respective parts (Lu et al., 2025).

To overcome the planning latency in deployment caused by the high-dimensional action space, we propose **Fac-TDMPC**, a novel extension of TDMPC that introduces **a factored world model** in the latent space. Instead of the densely connected neural networks used in TDMPC processing all action dimensions simultaneously, we employ factored dynamics, reward, and value functions for each action dimension $i$, following the factored decentralized Markov decision processes (fDec-MDPs) framework (Oliehoek & Amato, 2016). Given enough computational resources, this latent world model enables decentralized planning, reducing the optimization space to $\mathcal{O}(\max_i |\mathcal{A}^i|^H)$, where $\mathcal{A}^i$ denotes the action space of dimension $i$, thereby substantially reducing optimization complexity. To learn the factored world model, we employ a model distillation technique (Hinton et al., 2015) to transfer reward and value predictions from a centralized expert model. The objective of distilling models for planning is to make the student model preserve the expert model's prediction ordering over actions (Rusu et al., 2016), i.e., $f_s(s, a_1) < f_s(s, a_2)$ if $f_e(s, a_1) < f_e(s, a_2), \forall a_1, a_2 \in \mathcal{A}, s \in \mathcal{S}$, where $\mathcal{S}$ is the state space and $f_e, f_s$ are corresponding expert and student reward/value functions. We achieve this by incorporating Gaussian noise into action sequences and minimizing the KL divergence between the distributions predicted by their respective reward and value functions.

Beyond reducing planning latency, the factored latent model confers several practical advantages demonstrated in our experiments. (1) By enabling independent optimization per action dimension, Fac-TDMPC improves **robustness** to action perturbations: when individual actuators are disabled, or systematically scaled, the factored planner is less affected compared with the global planner. (2) The latent states and values of each agent can be separately visualized, which improves **interpretability** of the complete robotic system and helps to debug the failure cases. (3) Factored models naturally support **multi-task learning** across different robot embodiments: the shared joint-level structure allows data-efficient transfer, and in our multi-task learning experiments, Fac-TDMPC matches the centralized expert's average performance with a faster speed.

In summary, our contributions are as follows: (1) Fac-TDMPC—a novel MBRL framework that learns a factored world model to enable efficient planning in robotic tasks; (2) A model distillation procedure that transfers knowledge from a centralized expert world model (e.g., TDMPC) to a factored student world model preserving the prediction ordering over actions; (3) Empirical evidence showing that the proposed architecture achieves significant planning speedups, while maintaining the control performance and supporting robustness to action perturbations, explainability of robot behaviours and data-efficient multi-task learning.

## 2 Related Work

Model-based reinforcement learning (MBRL) (Moerland et al., 2023) integrates planning and learning by building a predictive model of the environment's dynamics and using it for decision-making, often achieving superior sample efficiency compared to model-free methods (Wang et al., 2019; Ye et al., 2021; Hansen et al., 2022; Hafner et al., 2023). Most existing approaches learn either reconstruction-based models (Hafner et al., 2023) or latent-space models with state abstractions (Hansen et al., 2022; 2023), which are used for planning at deployment. However, when unstructured latent-space models are combined with computationally heavy planners such as model predictive control (MPC), inference becomes slow, limiting real-time applicability in robotics. Our work, Fac-TDMPC, addresses this issue by learning factored world models and using decentralized planning to accelerate decision-making.

Our method requires learning a factored world model on which the latent state and action spaces are factored. The factorization idea has been explored in model-based planning settings. Some works (Balaji et al., 2021) assumed knowing the ground-truth factorization of the system before performing the planning process, which is limited to certain domains. Jiang et al. (2022) mapped the large action space into the compact latent space

and planned on that latent space, but required a complex variational encoder to reconstruct the original actions. In contrast, our method learns the factored world model in the latent space, without the prior knowledge of the system, and speeds up planning without action reconstruction.

Our paper utilizes a decentralized model predictive path integral (MPPI) (Anderson & Milutinović, 2013) algorithm to efficiently plan on the factored world model. More efficient planning methods designed for different special structures exist. Variable elimination (Guestrin et al., 2001) and max-plus (Kok & Vlassis, 2006) are exact and approximated decentralized planning methods on the coordination graph (Guestrin et al., 2001). Liu et al. (2023) adopted mixed-integer programming for sparsified neural networks. Dec-MCTS (Claes et al., 2017; Czechowski & Oliehoek, 2020) is similar to our setup that allows each robot to optimize its own actions, but with additional effort on estimating the team plan. Our method can be extended with these advanced planning methods when considering a more complex framework in the latent space. However, since we enforce the fDec-MDP framework in the latent space as introduced in Section 4, the decentralized MPPI is sufficient for efficient planning.

Our approach is also related to multi-agent reinforcement learning (MARL) (Bertsekas, 2019), where high-dimensional action spaces present similar challenges. In robotics, some methods decompose a single robot into multiple agents and apply MARL for control (Peng et al., 2021; Yan et al., 2024). However, these approaches require specifying factored state and action spaces *a priori*, which can be restrictive for generating the optimal actions. We instead learn the factored latent world model automatically, serving as an abstraction for efficient planning. In this work, we adopt reward and value decomposition from VDN (Sunehag et al., 2018) as a proof of concept, though more advanced MARL decompositions, such as QMIX (Rashid et al., 2018) or QTRAN (Son et al., 2019), could be integrated in future work.

## 3 Preliminaries

### 3.1 Factored Dec-MDP

A Markov decision process (MDP) (Bellman, 1957) is defined by the tuple $\langle \mathcal{S}, \mathcal{A}, T, R, \mu_0, \gamma \rangle$, where $\mathcal{S}$ and $\mathcal{A}$ denote the state and action spaces, $\Delta_{\mathcal{S}}$ and $\Delta_{\mathcal{A}}$ are the space of probability measure, $T : \mathcal{S} \times \mathcal{A} \to \Delta_{\mathcal{S}}$ is the transition function, $R : \mathcal{S} \times \mathcal{A} \to \mathbb{R}$ is the reward function, $\mu_0 \in \Delta_{\mathcal{S}}$ is the initial state distribution, and $\gamma \in [0, 1]$ is the discount factor. A factored decentralized MDP (fDec-MDP) (Oliehoek & Amato, 2016) is a special case of the MDP to model multi-agent systems, where $\mathcal{S} = \times_{i=1}^{N} \mathcal{S}^i$ and $\mathcal{A} = \times_{i=1}^{N} \mathcal{A}^i$ partition state and action spaces into $N$ disjoint subspaces. The fDec-MDP is said to be transition independent if $T(s_{h+1}|s_h, a_h) = \prod_{i=1}^{N} T_i(s_{h+1}^i|s_h^i, a_h^i)$, and reward independent if $R(s_h, a_h) = f(R_1(s_h^1, a_h^1) \dots R_N(s_h^N, a_h^N))$, where $f$ is a *monotonic function*: $\frac{\partial R}{\partial R_i} \geq 0, \forall i \in \{1, \dots, N\}, s_h^i, s_{h+1}^i \in \mathcal{S}^i, a_h^i \in \mathcal{A}^i$.

### 3.2 TDMPC

TDMPC (Hansen et al., 2022) falls into a class of model-based RL algorithms that learn a world model in the latent space (Schrittwieser et al., 2020; Hafner et al., 2023), which is aimed at facilitating real-time planning. Given that the robot receives image or proprioceptive state as global information $g_t$ at step $t$, it internally maintains an MDP in the latent space $\mathcal{S}$, which consists of the following components: (1) an encoder $s = f_E(g)$ maps observations to latent states; (2) a transition function $s' = f_T(s, a)$ models the dynamics in the latent space; (3) a reward function $\hat{r} = f_R(s, a)$ predicts the reward on the latent space; (4) a terminal value function $\hat{q} = f_Q(s, a)$ predicts the discounted sum of rewards, where $s, s' \in \mathcal{S}, a \in \mathcal{A}$.

TDMPC maintains a replay buffer $\mathcal{B}$ with collecting trajectories of horizon $H$ through interaction with the environment, and iteratively updates the world model using the data sampled from $\mathcal{B}$. The $f_E, f_T, f_R, f_Q$ components are jointly optimized to minimize the following objective function:

$$\mathcal{L}(f_E, f_T, f_R, f_Q) = \mathbb{E}_{(g,a,r)_{0:H+1} \sim \mathcal{B}} \Big[ \sum_{h=0}^{H} \lambda^h \Big( \|s_{h+1} - \text{sg}(f_E(g_{h+1}))\|_2^2 + \|\hat{r}_h - r_h\|_2^2 + \|\hat{q}_h - q_h\|_2^2 \Big) \Big], \quad (1)$$

where $s_0 = f_E(g_0), s_{h+1} = f_T(s_h, a_h), \hat{r}_h = f_R(s_h, a_h), \hat{q}_h = f_Q(s_h, a_h)$ are predictions of the world model. sg is the stop-gradient operation, $q_h = r_h + \max_a \gamma f_{\bar{Q}}(f_E(g_{t+1}), a)$ is the TD-target ($f_{\bar{Q}}$ is an exponential

moving-average version of $f_Q$), $\lambda \in [0, 1]$ is a hyperparameter to weigh less on further-step learning. TDMPC utilizes the learned world model by planning on the latent state space with model predictive path integral (MPPI) (Williams et al., 2016), a sampling-based MPC algorithm. Specifically, at each decision step $t$, the observation $g_t$ is first transformed into the corresponding latent state as $s_t = f_E(g_t)$, and MPPI is leveraged to efficiently find the maximized solution of $H$-step return $G^H(s_t, a_{t:t+H})$ as follows:

$$a_{t:t+H}^* = \underset{a_{t:t+H}}{\operatorname{argmax}} \quad G^H(g_t, a_{t:t+H}) = \sum_{h=t}^{t+H-1} \gamma^{h-t} f_R(s_h, a_h) + \gamma^H f_Q(s_{t+H}, a_{t+H}),$$
$$\text{s.t.} \quad s_t = f_E(g_t), s_{h+1} = f_T(s_h, a_h). \tag{2}$$

The optimal solution of step $t$'s action $a_t^*$ is acquired and executed in the environment. Optimizing Equation 2 is computationally expensive for tasks with continuous action spaces due to the argmax operation. MPPI therefore implements a sample-based approximation: it iteratively samples and evaluates a batch of candidate action sequences based on the previous proposal and refines the proposal by a weighted average (typically an importance-weighted average) of the higher-return samples. With sufficient samples per iteration and enough iterations, the proposal distribution concentrates on near-optimal trajectories of Equation 2.

## 4   Fac-TDMPC

In this section, we introduce **Fac-TDMPC**, a method designed to resolve the inference bottlenecks of traditional model predictive control by fundamentally restructuring the latent world model. Motivated by the parallelizability of multi-agent formulations, we project the global observation into a factored, transition-independent latent space governed by a factored decentralized Markov decision process (fDec-MDP). By distilling the value and reward distributions of a centralized expert into this factored model, Fac-TDMPC completely decentralizes the planning process across individual action dimensions. This architectural shift enables highly parallel, real-time control optimization without sacrificing the task performance of the original centralized framework.

### 4.1   A Motivating Example

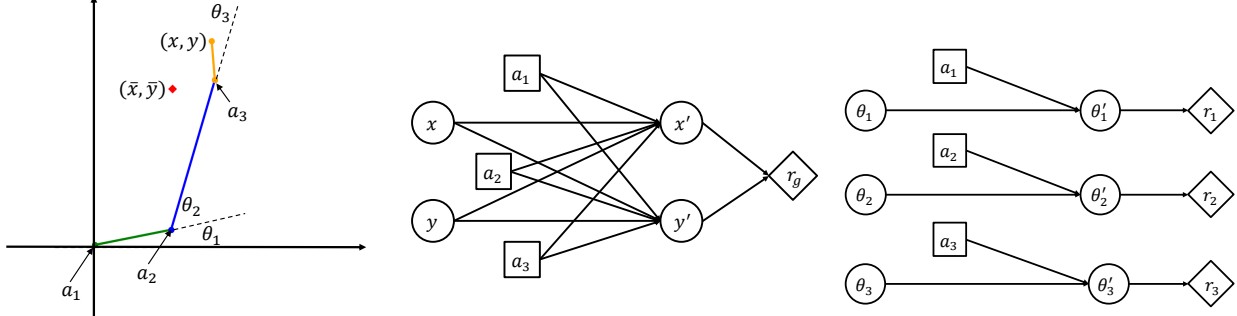

(a) Moving a 3-link 2-dimensional manipulator to the goal.

(b) DBN of a centralized MDP with the end-effector position as states.

(c) DBN of an fDec-MDP with the joint angles as states.

Figure 1: Different modelling methods to describe the task of moving a manipulator to the goal.

Before diving into our method, we first start with a simple manipulator task to motivate the use of the factored model. As illustrated in Figure 1a, the task is to move the end effector position $(x, y)$ of a 3-link manipulator to the goal position $(\bar{x}, \bar{y})$ on the 2-dimensional plane, with joint angles denoted as $\theta = (\theta_1, \theta_2, \theta_3)$. Notably, the end effector position and joint angles can be inter-converted with forward and inverse kinematics, written as $(x, y) = f_{FK}(\theta) = f_{FK}(f_{IK}(x, y))$. When the range of $(x, y, \theta)$ is restricted, such a transformation is bijective. The action $a = (a_1, a_2, a_3)$ consists of the forces put on the 3 joints. The joint force $a_i$ is first sent into a non-linear actuator model $\theta_i' = f_a(a_i, \theta_i)$ to achieve the next joint angle $\theta_i'$ and further influence the end effector's position.

One can simply use the end effector position $(x, y)$ as the state. Then the transition function becomes $(x', y') = f_{FK}(f_a(\theta_1, a_1), f_a(\theta_2, a_2), f_a(\theta_3, a_3))$, where $\theta = f_{IK}(x, y)$ and the reward function is $f_R(x, y, a) = -\|(x, y) - (\bar{x}, \bar{y})\|_2^2$. This centralized MDP model can be visualized as a dynamic Bayesian network (DBN) (Boutilier et al., 1999) in Figure 1b. Alternatively, one can view the joint angles as states, hence the problem can be formulated as an fDec-MDP with 3 agents. For each agent $i$, the transition function is simply the actuator model $\theta'_i = f_a(\theta_i, a_i)$, so as to be transition independent. The individual reward function $f_{R_i}(\theta_i, a_i) = -(\theta_i - \bar{\theta}_i)^2$, where $\bar{\theta} = f_{IK}(\bar{x}, \bar{y})$. When closed to the optimal point, the global reward function is monotonic on individual reward functions, written as $f_R(x, y, a) = M(f_{R_1}(\theta_1, a_1), f_{R_2}(\theta_2, a_2), f_{R_3}(\theta_3, a_3))$, where $M$ is a monotonic function (See Appendix A.1 for its detailed form).

Although these two modelling methods are equivalent to describe the system, planning using different models can result in huge differences. We apply the model predictive path integral control (MPPI) (Williams et al., 2016) on both models, but for fDec-MDP, each agent can be optimized independently and in parallel. Although both methods can reach the goal position in finite steps, the average planning time on the centralized MDP is 1.20 seconds per step (due to the expensive calculation of inverse and forward kinematics), while the average planning time on the fDec-MDP is 0.04 seconds per step, with around **30×** speedup (See Appendix A.2 for details). This example directly motivates us to incorporate the fDec-MDP modelling into the TDMPC framework for more efficient planning methods as introduced below.

## 4.2 Factored World Model

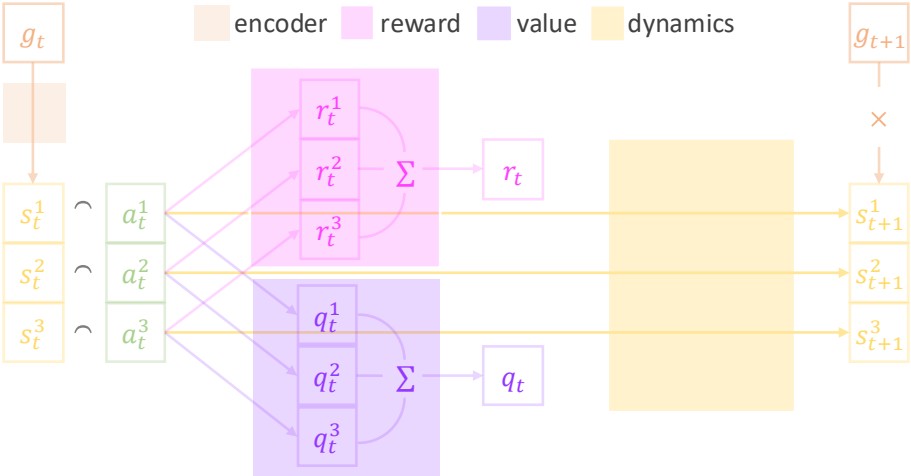

Figure 2: fDec-MDP modelling in the TDMPC framework. The observation $g_t$ is first encoded to the latent state $s_t$ by the encoder function $s_t = f_E(g_t)$. The latent state is factored to $N = \dim(\mathcal{A})$ agents as $s_t = \times_{i=1}^{N} s_t^i$. On each action dimension $i$, the independent dynamics function $f_{T_i}$, reward function $f_{R_i}$, value function $f_{Q_i}$ are defined locally. The global reward and value can be summed up by the local ones.

From Section 3.2, we observe that all planning in TDMPC is performed within the latent world model. **This motivates us to pose a research question: whether there exists a corresponding factored latent world model that supports optimal control, as the example in Section 4.1.** Such a factored world model is expected to be equivalent to the original in terms of generating optimal actions, but enables more efficient execution of planning algorithms such as MPPI.

To this end, we represent the latent world model as an fDec-MDP model (introduced in Section 3.1). We first view each action dimension of the robot as an independent agent, yielding $N = \dim(\mathcal{A})$ agents. Then, we design the factored latent space as $\mathcal{S} = \times_{i=1}^{N} \mathcal{S}_i : \mathbb{R}^{mN}$, where $m$ is the dimension for a single agent's hidden state. The encoder module is a centralized neural network as $s_t = f_E(g_t)$. In order to fully decompose agents for efficient planning, we further assume that the fDec-MDP is transition-independent so that for

each agent $i$, it maintains an independent local transition function $s_{t+1}^i = f_{T_i}(s_t^i, a_t^i)$. Meanwhile, each agent maintains an independent reward function and a value function, written as $r_t^i = f_{R_i}(s_t^i, a_t^i)$ and $q_t^i = f_{Q_i}(s_t^i, a_t^i)$. The global reward describing the system-wise performance is the sum of all individual rewards: $r_t = f_R(s_t, a_t) = \sum_{i=1}^N f_{R_i}(s_t^i, a_t^i)$. This condition is stronger than that of the example introduced in Section 4.1. The reward function in the example is strictly concave, so it is sufficient to ensure monotonicity only near the optimal point. However, since a general neural network is used here, we impose monotonicity of the global reward with respect to the individual rewards across all states $s$. As mentioned in Oliehoek & Amato (2016), transition independence and reward decomposition ensure that the value function is naturally decomposable, written as $q_t = f_Q(s_t, a_t) = \sum_{i=1}^N f_{Q_i}(s_t^i, a_t^i)$.

In summary, the complete factored world model consists of a centralized encoder function $f_E$, a factored transition function $f_T = (\times_{i=1}^N f_{T_i})$, a factored reward function $f_R = (\times_{i=1}^N f_{R_i})$, and the factored value function $f_Q = (\times_{i=1}^N f_{Q_i})$, as illustrated in Figure 2. During deployment, at each step $t$, the encoder $f_E$ first maps the global information $g_t$ to each agent's local state $s_t^i$. Based on this, each agent generates its own optimal action $a_t^{i*}$ using a planning algorithm $\pi_i$ (e.g., MPPI) and its learned factored world model $(f_{T_i}, f_{R_i}, f_{Q_i})$. The system-wide optimal action is then composed as $a_t^* = \times_{i=1}^N a_t^{i*} = \times_{i=1}^N \pi_i(s_t^i) = \times_{i=1}^N \pi_i(f_E(g_t)^i)$.

The latent state $s_t$ can be viewed as an abstraction of the global information $g_t$. Let the ground-truth reward and value function given the global state $g_t$ and action $a_t$ be denoted by $R(g_t, a_t)$ and $Q(g_t, a_t)$. Previous work (Li et al., 2006; Ni et al., 2023) categorized abstractions based on the information preserved in the latent state $s_t$ that supports different predictions. In our factored world model, these abstractions are represented as: (1) **$Q$-irrelevance abstraction** $(\phi_Q)$ for predicting the return, i.e. $Q(g_t, a_t) = f_Q(f_E(g_t), a_t) = \sum_{i=1}^N f_{Q_i}(s_t^i, a_t^i)$; (2) **reward prediction** (RP) for predicting the reward, i.e. $R(g_t, a_t) = f_R(f_E(g_t), a_t) = \sum_{i=1}^N f_{R_i}(s_t^i, a_t^i)$; (3) **next latent state prediction** (ZP) for predicting the next latent state, i.e. $s_{t+1} = f_E(g_{t+1}) = f_T(f_E(g_t), a_t) = \times_{i=1}^N f_{T_i}(s_t^i, a_t^i)$. Since we aim to learn an efficient world model that can equivalently generate optimal actions, we maintain abstractions in the level of $\phi_Q$ and RP, but not ZP which would impose a stricter but unnecessary level of abstraction (expressed as the cross mark between $g_{t+1}$ and $s_{t+1}$ in Figure 2). This relaxation broadens the applicability of the fDec-MDP framework from fully decomposable robotic tasks, as in the example, to more general robotic tasks—a claim further validated in our experiments.

## 4.3 Decentralized Planning

As one of the main benefits of adopting the factored model, we can further achieve an efficient planning method on the fly. Following the $H$-step return in TDMPC, we can similarly define an individual $H$-step return for agent $i$ with its individual dynamics, reward and value functions as $G_i^H(g_t, a_{t:t+H}^i) = \sum_{h=t}^{t+H-1} \gamma^{h-t} f_{R_i}(s_h^i, a_h^i) + \gamma^H f_{Q_i}(s_{t+H}^i, a_{t+H}^i)$, s.t. $s_t^i = f_E(g_t)^i$, $s_{h+1}^i = f_{T_i}(s_h^i, a_h^i)$, which is underpinned by the following proposition:

**Proposition 1 (Individual Global Max)** *For the fDec-MDP system with independent transition, reward and value functions, the $H$-step return $[G_i^H]$ satisfies the individual global max (Son et al., 2019) under observation $g_t$, which equivalently says:*

$$a_{t:t+H}^* = \underset{a_{t:t+H}}{argmax} \ G^H(g_t, a_{t:t+H}) = \times_{i=1}^N \underset{a_{t:t+H}^i}{argmax} \ G_i^H(g_t, a_{t:t+H}^i). \tag{3}$$

The proposition results from the additivity of the reward and value functions, and the independence of the dynamics function. It implies that each action dimension can be optimized in a decentralized manner for $N$ separate subsystems, without impairing the system-wise performance. For such a factored model, the optimization space of an MPC has been reduced from $\mathcal{O}(|\mathcal{A}|^H) = \mathcal{O}(\times_{i=1}^N |\mathcal{A}^i|^H)$ to $\mathcal{O}(\sum_{i=1}^N |\mathcal{A}^i|^H)$. Moreover, given sufficient computational resources, planning for each action dimension can be executed in parallel, yielding an additional $N$-fold speedup, with the time complexity $\mathcal{O}(\max_i |\mathcal{A}^i|^H)$.

### 4.4 Model Distillation

As introduced in Section 3.2, TDMPC employs a centralized world model $(f_{E_e}, f_{T_e}, f_{R_e}, f_{Q_e})$, which achieves strong task performance but is inefficient for planning. Our objective is to develop a factored model $(f_E, f_T, f_R, f_Q)$ that enables more efficient planning without sacrificing performance. To this end, we leverage model distillation (Hinton et al., 2015) to transfer knowledge from the centralized expert model of TDMPC. The distillation procedure is summarized in Algorithm 1.

---

**Algorithm 1** Model Distillation

---

1: **Input**: expert model $(f_{E_e}, f_{T_e}, f_{R_e}, f_{Q_e})$, dataset $\mathcal{B}$, hyperparameter $\lambda, H$
2: **Output**: factored model $(f_E, \times_{i=1}^{N} f_{T_i}, \times_{i=1}^{N} f_{R_i}, \times_{i=1}^{N} f_{Q_i})$
3: **for** iteration $i = 1, 2, ...., I$ **do**
4:      Sample a $H$-step sequential samples $(g, a)_{0:H}$ from the dataset $\mathcal{B}$
5:      Update parameters of the factored model based on the loss function as follows:
6:      $\mathcal{L}(f_E, f_T, f_R, f_Q) = \mathbb{E}_{(g,a)_{0:H} \sim \mathcal{B}} \left[ \sum_{h=0}^{H} \lambda^h \Big( D_{\mathrm{KL}}(\mathcal{R}_h, \mathcal{R}_h^e) + D_{\mathrm{KL}}(\mathcal{Q}_h, \mathcal{Q}_h^e) \Big) \right]$

---

The loss function in Line 6 aligns the reward and value *distributions* predicted by the factored model with those of the expert model via KL divergence. These objectives are introduced to ensure the $\phi_{Q^*}$ and RP level of state abstractions as introduced in Section 4.2. Following Rusu et al. (2016), we preserve the ordering of predicted actions rather than directly matching scalar values, since this has been shown to yield more effective knowledge transfer. Concretely, for a trajectory $(g, a)_{0:H}$ we inject Gaussian perturbations $\epsilon_{0:H} \sim \mathcal{N}^{H+1}(0, \Sigma)$ into actions and propagate them through the dynamics $s_{h+1} = f_T(s_h, a_h + \epsilon_h)$, yielding soft action distributions over rewards and values:

$$\mathcal{R}_h(a_h + \epsilon_h) \propto \exp\big(f_R(s_h, a_h + \epsilon_h)/T_s\big), \quad \mathcal{Q}_h(a_h + \epsilon_h) \propto \exp\big(f_Q(s_h, a_h + \epsilon_h)/T_s\big), \tag{4}$$

where $T_s$ is a temperature controlling the softness of the student predictions (Hinton et al., 2015). The expert distributions $\mathcal{R}_h^e$ and $\mathcal{Q}_h^e$ are defined analogously but with a different temperature $T_e$. We then minimize the KL divergence between student and expert distributions, which preserves the relative ranking of actions while allowing flexibility in scale. In practice, we approximate the KL terms with a small number of noise samples (10 in our experiments). Unlike the TDMPC objective (Equation 1), we omit explicit consistency losses, since our goal is a factored state abstraction that is optimized for planning rather than strict self-prediction (Li et al., 2006), as explained in Section 4.2.

## 5 Experiments

In this section, we present a comprehensive empirical evaluation of Fac-TDMPC to answer certain core research questions: (i) Can a transition-independent, factored latent world model preserve or enhance the control quality of a centralised expert across high-dimensional robotic tasks? (Section 5.2) (ii) To what extent does decentralized planning reduce wall-clock execution time and optimization sample complexity? (Section 5.3) (iii) Can this factored structure provide more explainability and robustness required in real-robot deployment? (Section 5.4 and 5.5) (iv) Do the learned factorized representations capture meaningful physical typologies and scale effectively to multi-task, cross-embodiment regimes? (Section 5.6)

### 5.1 Experimental Setup

**Task Setup.** We evaluate our method on DMControl (Tassa et al., 2018) benchmarks powered by the MuJoCo (Todorov et al., 2012) simulator. We select 7 robots with increasing state and action complexity as shown in Figure 3 following TDMPC (Hansen et al., 2022), but only robots with more than 3 action dimensions to validate the factored design, and meanwhile, we change the 2-link reacher to 3-link one; this yields 11 tasks in total (see Appendix B).

**Expert and Dataset Collection.** We train TDMPC with default hyperparameters (Hansen et al., 2022) on each task. The final checkpoint serves as the expert model, and its replay buffer of $10^6$ transitions is used as the offline dataset.

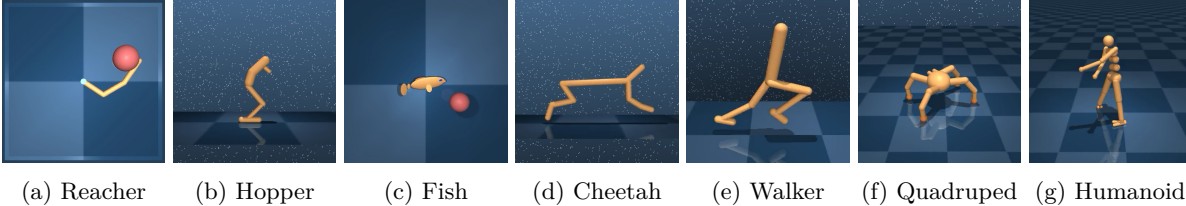

| (a) Reacher | (b) Hopper | (c) Fish | (d) Cheetah | (e) Walker | (f) Quadruped | (g) Humanoid |

Figure 3: Visualization of robots used in the experiments, with increased complexity in action space $\dim(\mathcal{A}) = \{3, 4, 5, 6, 6, 12, 21\}, \dim(\mathcal{O}) = \{8, 15, 24, 17, 24, 78, 67\}$.

**Baselines.** (1) **X% TDMPC** utilizes the fully centralized models as the expert model but with $X\%$ parameters, and trained with model distillation as introduced in Section 4.4. $X = 100, 10$ are set for the performance of the full and smaller expert models; (2) **TAP** (Zheng et al., 2020) reduces computational efficiency by transforming the original action space into a latent action space with VAE (Kingma & Welling, 2014) and planning in the latent space; (3) **Fac-TDMPC** is our proposed method as described in Section 4. A more detailed introduction on all baselines can be found in Appendix B.

**Training hyperparameters.** We perform model distillation (Section 4.4) for $10^5$ optimization steps and evaluate the factored model every $5 \times 10^3$ steps over 10 episodes. TDMPC and TAP's latent dimension is 512 in all experiments; Fac-TDMPC uses a per-agent latent size of $\lfloor 512/N \rfloor$ to keep parameter counts comparable. Other hyperparameters are shared across baselines and listed in Appendix B.3.

## 5.2 Training Performance

Figure 4 presents the training performance of all baselines. For nearly all tasks, Fac-TDMPC matches or exceeds the asymptotic performance of the expert model (TDMPC), demonstrating that the proposed factored architecture does not sacrifice control quality despite its reduced computational footprint (shown in the next section). In some challenging tasks, such as `Quadruped Walk` and `Humanoid Walk`, Fac-TDMPC even reaches superior performance than the expert model, which indicates the advantage of imitating the world model over simply behaviour cloning. 100% TDMPC attains similar performances as Fac-TDMPC and sometimes converges faster due to the same structure of the expert model. Notably, it fails the most challenging task, `Humanoid Walk`, showing the learning difficulty of such a large action space. In comparison, 10% TDMPC and TAP, with many fewer parameters, achieve reasonable performance in simple robots but cannot accommodate more complex ones. We do a detailed ablation study on key design choices of Fac-TDMPC in Section C.5.

We also evaluate Fac-TDMPC in an online model distillation scheme, where an expert model is trained from scratch following the online reinforcement learning process, and the student must simultaneously track the expert during the learning process. The results in Appendix C.1 illustrate that Fac-TDMPC can successfully track TDMPC in challenging tasks even when TDMPC is constantly updating. This finding is significant in that the efficient student model can be acquired simultaneously with the learning process of the full expert model, thereby saving training time.

## 5.3 Planning Efficiency

In this section, we validate the core motivation behind the proposed Fac-TDMPC: the learned factored structure enables more efficient downstream planning, in particular when applied with the MPPI algorithm (Williams et al., 2016). For MPPI, several key design choices critically affect planning efficiency: (i) the prediction time of the models, which determines the cost of a single run, and (ii) the number of samples and optimization iterations, which determine how many runs are required for precise planning.

Figure 5 compares the prediction time of all baseline models in robots with different action dimensions, since the Fac-TDMPC model structure relates only to the number of action dimensions. We only compare the speed of $(f_T, f_R, f_Q)$ without the encoder $f_E$ since all baselines apply the same size of feedforward network as encoders. All baselines are tested on a 1-thread CPU and Fac-TDMPC on $N$-thread CPU, depending

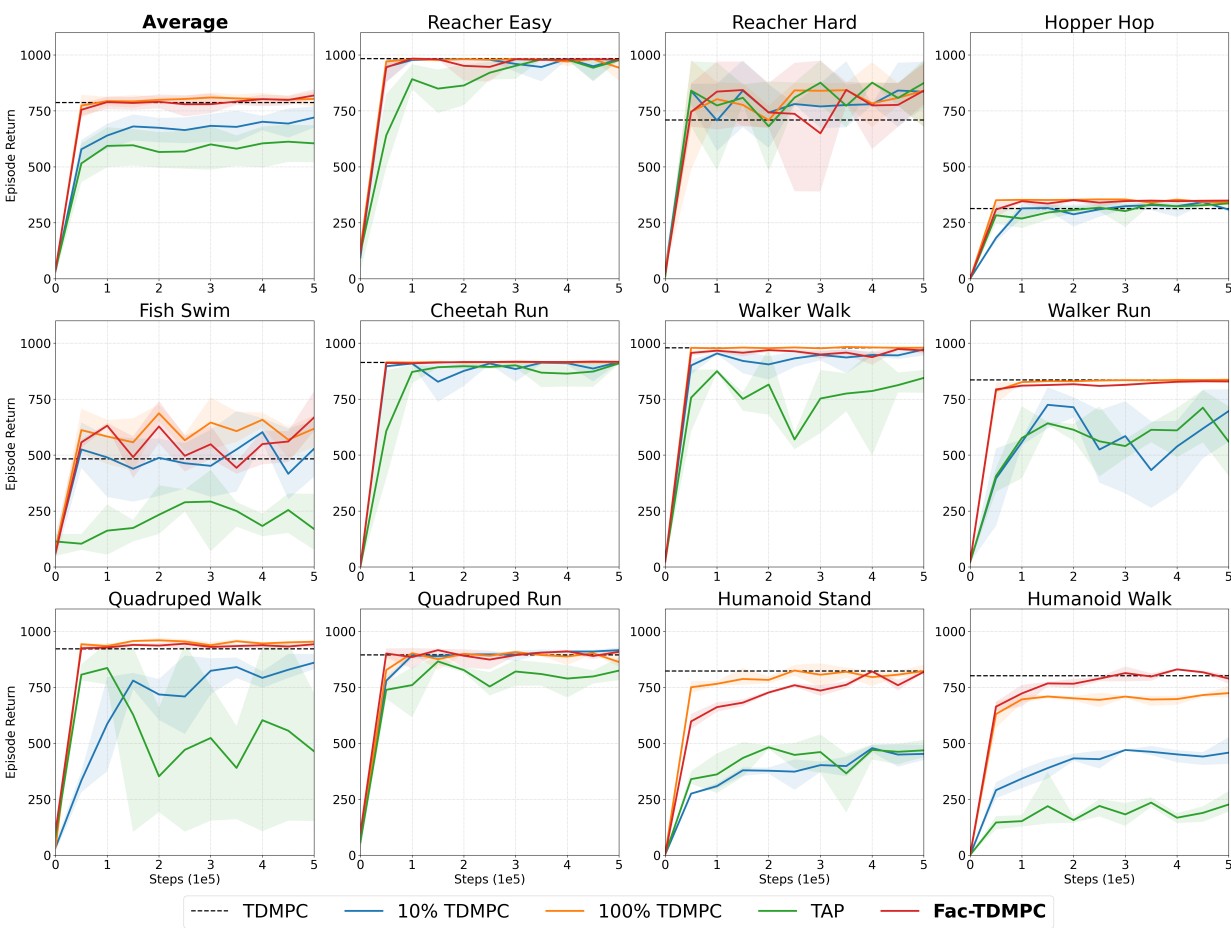

Figure 4: Training curves of all baselines on MuJoCo tasks. Each curve shows the average episode return over three seeds, with shaded regions denoting the standard deviation.

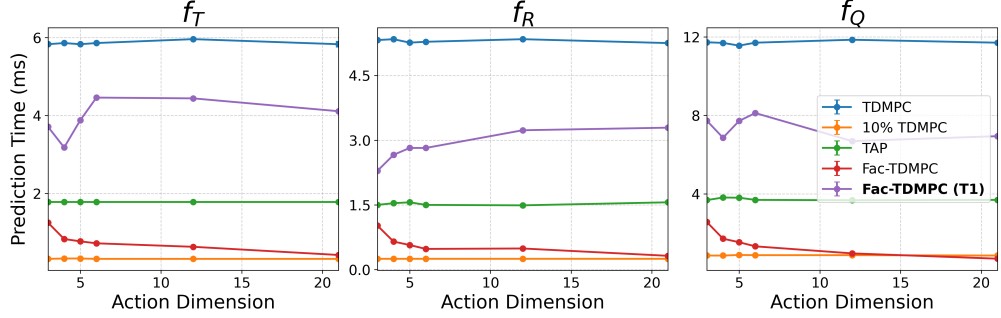

Figure 5: Prediction time of different models on different robots. **All wall times are reported in milliseconds** with average and standard deviation over 100 runs.

on the robot's action dimensions $N$, to enable parallel prediction. Notably, the prediction time for $X\%$ TDMPC and TAP are similar across different robots, since the model structures are unchanged. Across all robots and model components, Fac-TDMPC achieves a substantial reduction in per-step computation time compared to TDMPC, and the efficiency gain continues to improve with the increase in action dimensions. Furthermore, we evaluate the inference time of Fac-TDMPC in a single-threaded setting ("T1"), where it exhibits computational requirements comparable to the TDMPC baseline. This observation confirms that the

observed speedup is primarily driven by the parallelized prediction capabilities of the factored architecture. Specifically, for an action space of dimension $N$, a baseline TDMPC model of size $\mathcal{O}(K)$ is decomposed into $N$ individual sub-models, each with a size of approximately $\mathcal{O}(K/\sqrt{N})$ to maintain a comparable total parameter count. This factored structure facilitates more efficient computation on a single-threaded CPU and further accelerates the optimization and inference processes when distributed across $N$ independent threads, thereby enabling significant wall-clock efficiency through parallel computation. Its speed even matches 10% TDMPC for humanoid robots, but the control performance is much better as shown in Section 5.2.

Prediction time determines the cost of a single call of the models, while the number of calls also matters. In MPPI, more samples and iterations implies more calls to the models. We test the control performance of the `Walker Run` task with various numbers of samples and iterations, which is reported in Figure 6. The default numbers of samples and iterations are 512 and 6. Reducing them will undermine the control performance of both TDMPC and Fac-TDMPC. However, the Fac-TDMPC suffers a much smaller reduction, and even achieves acceptable performance for 128 samples or 2 iterations, which indicates the possibility of reducing computational cost by $3 \sim 4\times$ without sacrificing planning performance. We attribute this to the individual global max property of the return (Proposition 1), which reduces the optimization space from $\mathcal{O}(\times_{i=1}^{N}|\mathcal{A}^i|^H)$ to $\mathcal{O}(\sum_{i=1}^{N}|\mathcal{A}^i|^H)$. This leads the factored world model to better utilization of limited optimization resources, which is crucial in robotic tasks (Duan et al., 2025).

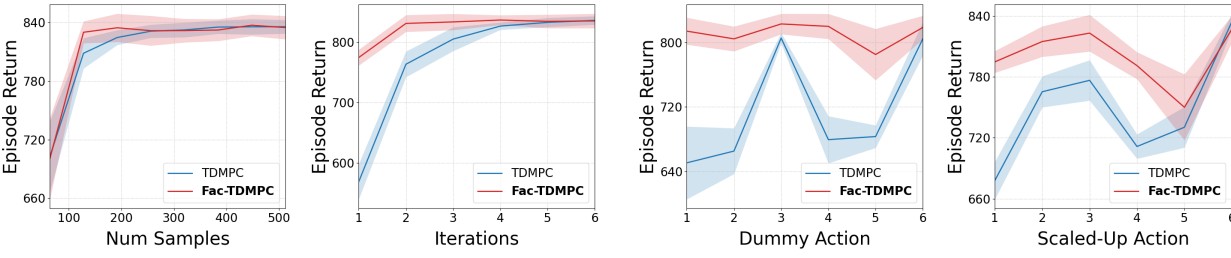

Figure 6: Control performance on `Walker Run` with various optimization parameters.

Figure 7: Control performance on `Walker Run` under various perturbations.

### 5.4 Robustness to Action Perturbations

The factored structure permits independent optimization of each action dimension that can improve robustness to action perturbations. We evaluate two perturbations on the `Walker Run` task: (1) one action dimension is *dummy* that always follows a Normal distribution; (2) one action dimension is consistently scaled, simulating modified motor strength. Results in Figure 7 show that certain action dimensions are more sensitive to the perturbations, and Fac-TDMPC maintains more stable and higher episode returns across perturbed dimensions.

### 5.5 Visualization on Factored World Model

In this section, we visualize the learned Fac-TDMPC world model to closely explain the factored behaviours. We select the 3-link reacher robot as an example, and the goal is to reach the red point in the 2-dimensional planar space. The global reward is decided by the distance between the goal position and the end-effector position. For Fac-TDMPC, 3 agents are collaborating to control the joints and finish the task.

We first visualize the latent states of all agents collected from 100 episodes. Each agent's latent state is a 50-dimensional vector; we project these vectors into two dimensions using t-SNE (van der Maaten & Hinton, 2008), as shown in Figure 8. The projection reveals distinct clusters corresponding to different joints. Interestingly, these three clusters exhibit a degree of rotational invariance, which implies that the individual latent states also encode some common knowledge of the entire system for further coordination, cf. the joints' local views of the robot's global states (Yan et al., 2024). We visualize individual and global $H$-step returns for three representative episodes in Figure 8, and include two rendered snapshots (top) corresponding to the

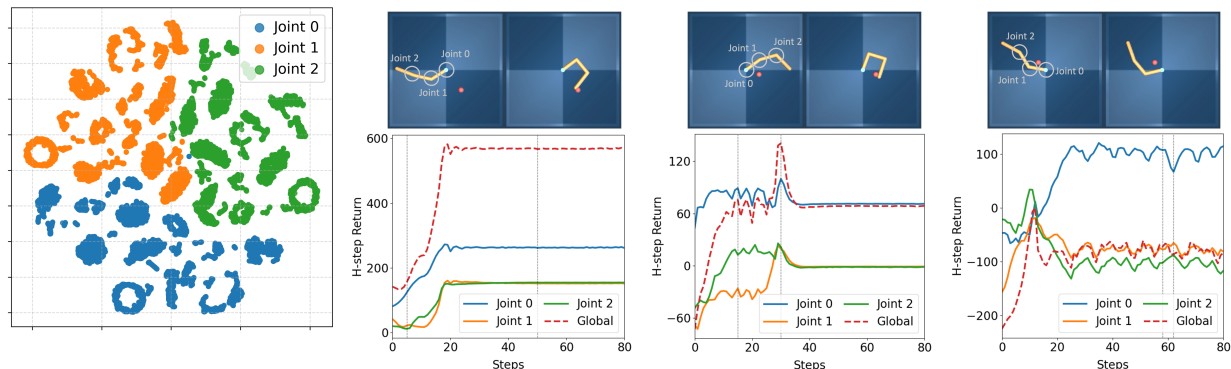

Figure 8: Visualization of latent states and credit assignment. **Left:** t-SNE projection of latent states shows clear clustering by joints. **Middle and right:** individual and global H-step returns of different episodes. The upper two rendered images correspond to the steps with vertical dashed lines in the plots.

steps marked by vertical dashed lines in the plots. The first two illustrate successful coordination, while the third shows a failure case. In the first success, all joints undergo large angular changes to reach the target position, leading to consistent increases in both individual and global returns. In the second success, the phase shift is primarily driven by Joint 1, which accounts for the sharp rise in the orange and red curves. In contrast, the failure case exhibits oscillatory patterns across all curves, where the efforts of Joint 0 conflict with those of Joints 1 and 2. Among these states, the additive reward and value functions are insufficient to capture the global behaviours. Addressing such cases may require more advanced coordination strategies, such as Q-MIX (Rashid et al., 2018). In general, visualizations of latent states and returns provide a powerful tool for analyzing robotic systems and explaining their behaviours.

## 5.6 Multi-task Learning

As a final contribution, we investigate the performance of Fac-TDMPC in a multi-task learning scenario. We adopt the multi-task benchmark introduced in Hansen et al. (2023), which aggregates 30 tasks from the DeepMind control suite (Tassa et al., 2018). These tasks span 10 robot embodiments with action dimensionalities ranging from 1 to 6. The dataset contains trajectories collected by a 5M-step TDMPC2 expert policy (Hansen et al., 2023), resulting in approximately $3.5 \times 10^8$ offline transitions. The expert achieves an average episode return of 283.0 across the tasks.

To apply Fac-TDMPC in this setting, we train a *single shared factorized world model* across all embodiments and tasks. For a robot with $N$ actuated joints, the latent state $s_t$ is decomposed into $N$ joint-specific latent factors, $s_t = \{s_t^1, \ldots, s_t^N\}$, where each factor corresponds to one action dimension (joint). To capture task-specific variations such as robot morphology or control objectives, we follow Hansen et al. (2023) and introduce a learned task embedding $e_k$ for each task $k$, which is further concatenated to each factor's latent state to an augmented latent state $\bar{s}_t = \{(s_t^1, e_k), \ldots, (s_t^N, e_k)\}$. Each joint factor is modeled using a shared set of networks that parameterize its latent dynamics, reward and value functions $(f_{T_i}, f_{R_i}, f_{Q_i})$. The dynamics prediction for task $k$'s joint $i$ is represented as $s_{t+1}^{(i)} = f_{T_i}(s_t^i, a_t^i, e_k)$. Since the shape of the augmented factored latent state $(s_t^i, e_k)$ is fixed , the factored world model can be shared *shared across tasks*, allowing the model to reuse joint-level knowledge across different robot embodiments. The world model and policy are trained jointly on the pooled offline dataset following the model distillation training procedure as in Section 5.1.

The leftmost plot of Figure 9 shows that Fac-TDMPC surpasses the performance of the expert policy while exhibiting significantly improved data efficiency compared to fully centralized models, particularly during the early stages of training. To further analyze the learned representation, we visualize the latent states of the walker and hopper robots. The hopper can be interpreted as a one-legged variant of the walker with an additional waist joint. The middle plot of Figure 9 shows the projected latent states for three tasks involving the walker robot. The resulting embeddings form distinct clusters corresponding to individual joints, and

these clusters remain consistent across tasks with the same embodiment. The second plot from the right illustrates cross-embodiment knowledge transfer between the walker and hopper robots. We observe weak but consistent correlations between the walker's joints 0–2 and the hopper's joints 1–3 in the projected latent space. These joints correspond to the hip, knee, and ankle of one leg in both embodiments. This alignment indicates that the factorized representation captures shared joint-level dynamics across different robots, which helps explain the improved data efficiency of Fac-TDMPC.

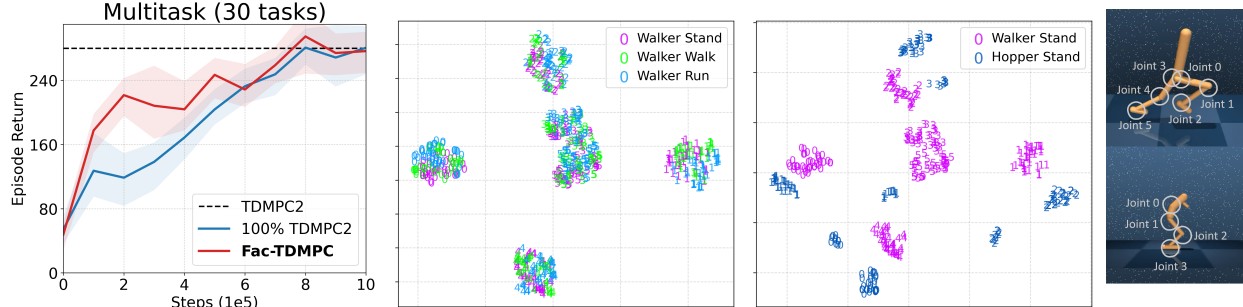

Figure 9: Multi-task learning results. **Left:** Training curves of multi-task learning. The episode return is averaged across 30 tasks. **Middle and right:** t-SNE projection of latent states both within robots (walker) and between robots (walker and hopper). The sign of the points indicates the actual joint index.

## 6 Conclusion

This paper introduces Fac-TDMPC, a factored world model in the latent space, enabling fast decision-making on deployment and robustness for high-dimensional action scenarios. Fac-TDMPC achieves significant speedups in planning through decentralized planning while maintaining high control performance on various robotic tasks in MuJoCo. Alongside, the learned factored structure enhances robustness to action perturbations.

Notably, in a few highly dynamic tasks that require tight and global coordination, Fac-TDMPC can learn more slowly than a centralized world model. To address these limitations we plan several extensions in future work: (1) incorporate latent histories or recurrent encoders to capture temporal coupling and better model interactions between dimensions (Oliehoek et al., 2021); (2) integrate advanced credit-assignment mechanisms (e.g., QMIX (Rashid et al., 2018) or QTRAN (Son et al., 2019) factorization) to better enforce cooperative behaviors; (3) explore hybrid or learned coupling modules and attention-based factor discovery to automatically and dynamically decompose the action space rather than assuming fixed independence.

**Acknowledgments**

This research received funding from the European Union's Horizon 2020 research and innovation program under the Marie Skłodowska-Curie grant agreement No. 953348 (ELO-X). This work was part of BrainLinks-BrainTools, which was funded by the Federal Ministry of Economics, Science and Arts of Baden-Württemberg within the sustainability program for projects of the excellence initiative II. Jianhong Wang is supported by the Engineering and Physical Sciences Research Council (EPSRC) [Grant Ref: EP/Y028732/1].

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

## Appendix

## A Additional Method Details

### A.1 Monotonic Function for the 3-Link Manipulator

Let $r_g = f_R(x, y, a) = -\|(x, y) - (\bar{x}, \bar{y})\|_2^2$ the global reward and $r_i = f_{R_i}(\theta_i, a_i) = -(\theta_i - \bar{\theta}_i)^2$ the individual rewards. We are going to show $r_g = M(r_1, r_2, r_3)$ when $(x, y) \to (\bar{x}, \bar{y})$, where $M$ is a monotonic function. The specific format of the forward kinematics in our case is $(x, y) = f_{FK}(\theta_1, \theta_2, \theta_3) = (L_1 \cos\theta_1 + L_2 \cos(\theta_1 + \theta_2) + L_3 \cos(\theta_1 + \theta_2 + \theta_3), L_1 \sin\theta_1 + L_2 \sin(\theta_1 + \theta_2) + L_3 \sin(\theta_1 + \theta_2 + \theta_3))$.

We first represent joints $\theta$ by individual rewards:

$$r_i = -(\theta_i - \bar{\theta}_i)^2 \tag{5}$$

$$\theta_i = \bar{\theta}_i \pm \sqrt{-r_i} \tag{6}$$

Then replace all joints in the global reward by individual rewards:

$$r_e = -\|(x, y) - (\bar{x}, \bar{y})\|_2^2 \tag{7}$$

$$= -\|f_{FK}(\theta) - (\bar{x}, \bar{y})\|_2^2 \tag{8}$$

$$= -\|f_{FK}(\bar{\theta}_1 \pm \sqrt{-r_1}, \bar{\theta}_2 \pm \sqrt{-r_2}, \bar{\theta}_3 \pm \sqrt{-r_3}) - (\bar{x}, \bar{y})\|_2^2 \tag{9}$$

$$= M(r_1, r_2, r_3) \tag{10}$$

We now have the relations between global reward and individual, we are going to show that $M$ is monotonic when $(x, y) \to (\bar{x}, \bar{y})$ or $\theta \to \bar{\theta}$. Let $c_e = -r_e$, $c_i = \bar{\theta}_i \pm \sqrt{-r_i}$, and the relation becomes as follows:

$$c_e = \|f_{FK}(c_1, c_2, c_3) - (\bar{x}, \bar{y})\|_2^2 \tag{11}$$

$$= (L_1 \cos c_1 + L_2 \cos(c_1 + c_2) + L_3 \cos(c_1 + c_2 + c_3) - \bar{x})^2 \tag{12}$$

$$+ (L_1 \sin c_1 + L_2 \sin(c_1 + c_2) + L_3 \sin(c_1 + c_2 + c_3) - \bar{y})^2 \tag{13}$$

$$\frac{\partial c_e}{\partial c_3} = 2L_3[(y - \bar{y}) \cos(c_1 + c_2 + c_3) - (x - \bar{x}) \sin(c_1 + c_2 + c_3)] \tag{14}$$

$$\frac{\partial c_e}{\partial c_2} = 2[(y - \bar{y})(L_2 \cos(c_1 + c_2) + L_3 \cos(c_1 + c_2 + c_3)) - (x - \bar{x})(L_2 \sin(c1 + c_2) + L_3 \sin(c_1 + c_2 + c_3)] \tag{15}$$

$$\frac{\partial c_e}{\partial c_1} = 2[(y - \bar{y})(L_1 \cos c_1 + L_2 \cos(c_1 + c_2) + L_3 \cos(c_1 + c_2 + c_3)) \tag{16}$$

$$- (x - \bar{x})(L_1 \sin c_1 + L_2 \sin(c_1 + c_2) + L_3 \sin(c_1 + c_2 + c_3)] \tag{17}$$

$$= 2[(y - \bar{y})x - (x - \bar{x})y] = 2(y\bar{x} - x\bar{y}) \tag{18}$$

We can focus on the case of $c_i = \bar{\theta}_i + \sqrt{-r_i}$, and then $\bar{x} - x = \sqrt{c_e} \sin(c_1 + c_2 + c_3) > 0, y - \bar{y} = \sqrt{c_e} \cos(c_1 + c_2 + c_3) > 0$. And otother cases followin the same procedure.

$$\frac{\partial c_e}{\partial c_3} = 2L_3[\sqrt{c_e} \cos^2(c_1 + c_2 + c_3) + \sqrt{c_e} \sin^2(c_1 + c_2 + c_3)] = 2L_3\sqrt{c_e} > 0 \tag{19}$$

$$\frac{\partial c_e}{\partial c_2} = 2[L_3\sqrt{c_e} \cos^2(c_1 + c_2 + c_3) + L_2L_3 \cos(c_1 + c_2) \cos(c_1 + c_2 + c_3)) \tag{20}$$

$$+ L_3\sqrt{c_e} \sin^2(c_1 + c_2 + c_3) + L_2L_3 \sin(c1 + c_2) \sin(c_1 + c_2 + c_3)] \tag{21}$$

$$= 2[L_3\sqrt{c_e} + L_2L_3 \cos c_3] > 0 \tag{22}$$

$$\frac{\partial c_e}{\partial c_1} = 2(y\bar{x} - x\bar{y}) > 0 \tag{23}$$

In addition, we know that $\frac{\partial c_e}{\partial r_e} < 0$ and $\frac{\partial c_i}{\partial r_i} < 0$. By the transitivity of monotonicity, $\frac{\partial r_e}{\partial r_i} > 0$, so that $M$ is a monotonic function.

### A.2   Experiments for the 3-Link Manipulator

**Hyperparameters of MPPI.** We apply the model predictive path integral control (MPPI) (Williams et al., 2016) on both centralized MDP and fDec-MDP. In specific, the shared hyperparameters of MPPI are shown in Table 1 for fair comparison. Notably, the discount is set to 0.0 for simplicity, so that only transition and reward functions matter in our cases. For fDec-MDP, since each agent (joint) has individual transition and reward functions, the optimization can be executed independently and in parallel.

Table 1: Hyperparameters of MPPI for the 3-Link Manipulator.

| Planning (MPPI / MPC) | | |
|---|---|---|
| iterations | 6 | Number of MPPI optimization iterations per control step. |
| num_samples | 50 | Number of trajectory samples drawn per iteration. |
| num_elites | 5 | Number of top samples (elites) used to update the sampling distribution. |
| horizon | 4 | Planning horizon (timesteps) for MPPI (comment shows prior value 5). |
| min_std | 0.05 | Minimum standard deviation for the sampling noise (stability floor). |
| max_std | 2.0 | Maximum standard deviation allowed for sampling noise. |
| temperature | 0.5 | Softmax/temperature parameter used when computing weights for samples. |
| discount | 0.0 | Discount factor. |
| freq | 10 | Planning frequency, the step interval is then 0.1 seconds. |
| steps | 100 | Planning steps. |

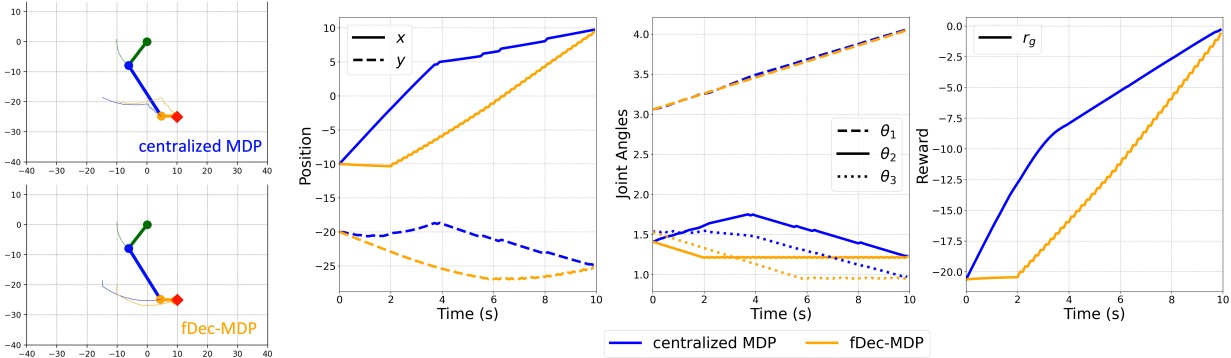

Figure 10: Planning results on both centralized MDP and fDec-MDP. **Left:** The trajectory and the final state of both methods. The trajectory of all joints are visualized in solid lines. **Middle and right:** The x-axis is the elapsed time, and the y-axis represents the end effector position, joint angles and total reward.

**Results.**   We visualize the two planning results in Figure 10. Both methods reach the goal position in the end as shown by the reward, and planning on the fDec-MDP leads to a smoother trajectory on the end effector's positions. Besides, distributed planning on fDec-MDP drives $\theta_2$ and $\theta_3$ more actively for faster convergence, instead of being predominantly occupied by one dimension ($\theta_1$). Both planning processes are executed on the same hardware and hyperparameters. The average planning time on the centralized MDP is 1.20seconds per step, while the average planning time on the fDec-MDP is 0.04 seconds per step, with around ×**30** speedup.

### A.3   Violation on Individual Global Max

The success of decentralized planning method depends on the existence of Proposition 1 as introduced in Section 4.3, i.e. $a^*_{t:t+H} = \underset{a_{t:t+H}}{\mathrm{argmax}}\ G^H(g_t, a_{t:t+H}) = \times_{i=1}^{N} \underset{a^i_{t:t+H}}{\mathrm{argmax}}\ G^H_i(g_t, a^i_{t:t+H})$. In other words, if finding the decomposition of the global returns into the sum of the individual returns as $G^H(g_t, a_{t:t+H}) = \sum_i G^H_i(g_t, a^i_{t:t+H})$ either theoretically (as the 3-link manipulator example in Section 5) or empirically (as the MuJoCo experiments in Section 5), one can say in confidence that the decentralized planning provdes

the exact same results as the global planning method. Here, we provide a failure case that does not satisfy the above-mentioned decomposition. Assume a two-agent case with horizon $H = 1$ and constant states $g_t = g$ for simplicity, we can abbreviate the condition as $G(a) = G_1(a^1) + G_2(a^2)$. For the global return with structure: $G(a) = a^1a^2$, it is not hard to prove that we can never find $G_1, G_2$ to exactly satisfy the condition $G(a) = G_1(a^1) + G_2(a^2)$. When applied to robotics, a perfect example is a robot using two arms to lift a heavy object. Only when both arms lift with equal force can the task succeed. It is difficult to find a decomposition to execute decentralized planning in this scenario even with the global state $g$ for both agents.

### A.4  Deviations from full factorization and optimality

This problem can be answered from the perspective of how the terminal value learning in our algorithm is updated in function approximation version. Basically, the update of each decentralized terminal value function $f_{Q_i}^{\theta_i}(s_t^i, a_t^i)$ with stochastic gradient descent (SGD) as the optimizer follows the equation:

$$\theta_i \leftarrow \theta_i + \alpha \left( r_t + \max_a \gamma f_{\bar{Q}}(s_{t+1}, a) - \sum_{j=1}^{N} f_{Q_j}^{\theta_j}(s_t^j, a_t^j) \right) \nabla_{\theta_i} f_{Q_i}^{\theta_i}(s_t^i, a_t^i), \tag{24}$$

where each $f_{Q_i}^{\theta_i}(s_t^i, a_t^i)$ is presumed to be parameterized by $\theta_i$, $f_{\bar{Q}}$ functions the global terminate value, and $\alpha$ is the learning rate. It is obvious that the approximation error induced by the full factorization would influence the credit denoted as $r_t + \max_a \gamma f_{\bar{Q}}(s_{t+1}, a) - \sum_{j=1}^{N} f_{Q_j}^{\theta_j}(s_t^j, a_t^j)$, during the optimization process of updating $\theta_i$. Assume this error is $\epsilon_t$ at some step $t$, the credit for updating parameters would be deviated by $\alpha\epsilon_t$. As a result, it will lead to total error of the resulting parameter, accumulated along the optimization trajectory, upper bounded by $m\alpha\epsilon_{\max} \max_{s_t^i, a_t^i} \nabla_{\theta_i} f_{Q_i}^{\theta_i}(s_t^i, a_t^i)$, where $m$ is the number of updating parameters and $\epsilon_{\max} = \max_t \epsilon_t$.

### A.5  Discussion on modular world models

There are also works learning structured world models in the latent space. One choice is to learn a factored world model in the latent space (Balaji et al., 2021; Lei et al., 2023), which usually aims to capture the causality of the system with attention-based architectures. However, directly using it as the transition function in the MPPI method does not incur efficient planning. Because the factored structure is still a connected graph, and the optimization space is still $\mathcal{O}(|\mathcal{A}|^H)$, in comparison with Fac-TDMPC $\mathcal{O}(\sum_{i=1}^{N} |\mathcal{A}^i|^H)$. Among the literature, Liu et al. (2023) is motivated to learn a structured transition function for further efficient planning. They first learned a sparsified neural network and then adopted mixed-integer programming (MIP) to plan on it. However, as is well known, MIP planner is usually slower than MPPI planner, and their methods are restricted to the ReLU activations in the transition functions. Our method allows arbitrary forms of transition functions.

## B  Additional Experimental Setups

### B.1  Task Description

- `Reacher Easy`: A simple reaching task where a 3-link arm must move its end-effector to a target position with relatively slow dynamics.

- `Reacher Hard`: A more challenging version of the 3-link reaching task with faster dynamics or smaller target tolerance, requiring more precise control.

- `Hopper Hop`: A single-legged hopper must maintain balance and hop forward, testing stability and rhythmic control.

- `Fish Swim`: A simulated fish must propel itself forward and maneuver in water, involving complex body coordination.

- `Cheetah Run`: A two-legged cheetah to run while keeping stability under high-speed conditions.

- `Walker Walk`: A bipedal walker must move forward at a moderate speed, emphasizing stable gait generation.

- `Walker Run`: The walker must achieve faster locomotion while maintaining balance, increasing control difficulty.

- `Quadruped Walk`: A four-legged robot must walk steadily forward, testing coordination across multiple legs.

- `Quadruped Run`: A faster variant requiring the quadruped to run while keeping stability under high-speed conditions.

- `Humanoid Stand`: A two-legged humanoid robot must stand steadily, testing coordination across multiple joints.

- `Humanoid Walk`: A faster variant requiring the humanoid to walk while keeping stability.

### B.2  Baselines

For TDMPC, we directly adopt the official source code[1] as the implementation. For 10% TDMPC, we directly set the hidden units from 512 to 66, ending up with around 10% parameters of the original model. For TAP, the original method learns a latent action space, compressing both large action space and long time horizons. To fairly compare with other methods in the following MPPI planning methods, we restrict the compression only in the action space. As a result, the method essentially becomes to map the original action space into a latent action space, empirically set as 2 dimensions. Both transition, reward and value functions are grounded on the latent state and action space. The same planning algorithm MPPI is performed on the latent action space, and mapped to the original action space with a trained decoder. The training is accomplished with an additional VAE reconstruction loss on actions.

### B.3  Hyperparameters

To compare all algorithms fairly, we set the model structures and hyperparameters equally. All algorithms are trained with Adam optimizer (Kingma & Ba, 2015). The full hyperparameters are shown in Table 2. All experiments are carried out on NVIDIA GeForce RTX 2080 Ti and Pytorch 1.10.1.

## C  Additional Experimental Results

### C.1  Online Model Distillation

We further evaluate Fac-TDMPC in an online model distillation setting, where the lightweight student model is required to continuously track the expert policy as it improves during training. In this setting, the expert (TDMPC) is not fixed but rather being optimized from scratch and in parallel, posing a more challenging scenario than standard offline distillation.

Figure 11 summarizes the results across 3 most challenging tasks (`Walker Run`, `Quadruped Run`, and `Humanoid Walk`) as well as the averaged performance. Despite the non-stationarity introduced by the evolving expert, Fac-TDMPC is able to reliably follow the learning curve of TDMPC across all tasks. This finding is particularly meaningful: it shows that an efficient distilled model can be acquired online, simultaneously with the training of the full expert model. Such a property eliminates the need for a separate post-hoc distillation phase, thereby reducing overall training time and enabling practical deployment of efficient policies in real-time learning scenarios.

The result for `Humanoid Walk` is interesting, in that Fac-TDMPC even surpasses TDMPC at the beginning, since learning a centralized model is difficult for such a large action space. But in the end, the minor performance gaps that appear might result from a slow convergence of the expert model.

---

[1]https://github.com/nicklashansen/tdmpc

Table 2: Hyperparameters used in experiments.

| Training | Values | Description |
|---|---|---|
| steps | 10_000_00 | Total number of training steps (gradient updates). |
| batch_size | 512 | Mini-batch size used for each optimization step. |
| reward_coef | 0.5 | Weight for reward prediction / reward-loss term in the training objective. |
| value_coef | 0.1 | Weight for value / critic loss term. |
| consistency_coef | 0.0 | Weight for any model-consistency loss (if used). |
| rho | 0.5 | Mixing / interpolation coefficient used in targets or blending (implementation-specific). |
| lr | 1e-4 | Base learning rate for optimizer. |
| enc_lr_scale | 1.0 | Multiplier applied to the encoder learning rate (encoder lr = lr × this). |
| grad_clip_norm | 10 | Maximum gradient norm for clipping. |
| tau | 0.01 | Soft-update coefficient for target networks (EMA factor). |
| discount | 0.99 | Discount factor. |
| buffer_size | 1_000_000 | Replay buffer capacity (number of transitions). |
| **Planning (MPPI / MPC)** | | |
| mpc | true | Whether to enable model-predictive control (MPPI) at inference time. |
| iterations | 6 | Number of MPPI optimization iterations per control step. |
| num_samples | 512 | Number of trajectory samples drawn per iteration. |
| num_elites | 64 | Number of top samples (elites) used to update the sampling distribution. |
| num_pi_trajs | 24 | Number of policy (pi) trajectories evaluated (if using policy proposals). |
| horizon | 3 | Planning horizon (timesteps) for MPPI (comment shows prior value 5). |
| min_std | 0.05 | Minimum standard deviation for the sampling noise (stability floor). |
| max_std | 2.0 | Maximum standard deviation allowed for sampling noise. |
| temperature | 0.5 | Softmax/temperature parameter used when computing weights for samples. |
| **Architecture** | | |
| num_enc_layers | 2 | Number of encoder (feedforward/conv) layers. |
| enc_dim | 256 | Width (hidden units) of encoder layers. |
| num_channels | 32 | Number of channels if using convolutional layers (per layer). |
| mlp_dim | 512 | Width of MLP heads / fully-connected layers. |
| latent_dim | 50 | Dimension of learned latent state per agent (or total, per design). |
| num_q | 2 | Number of Q-networks (ensemble) used for critic. |
| dropout | 0.01 | Dropout probability used in networks. |

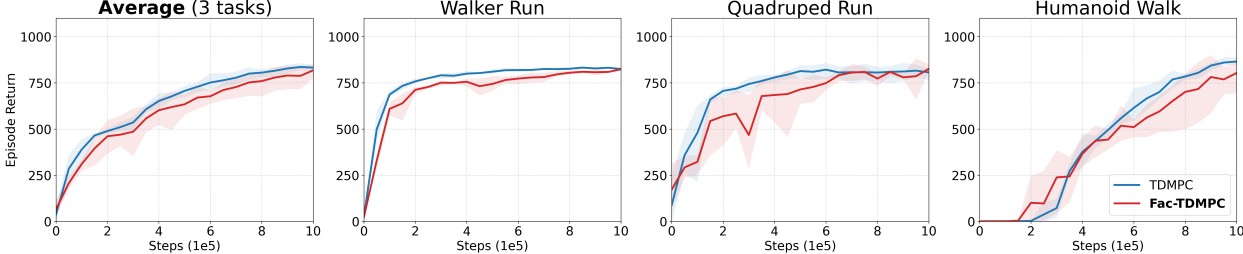

Figure 11: Online model distillation on challenging (`Walker Run`, `Quadruped Run`, and `Humanoid Walk`) tasks. Each curve shows the average episode return over three seeds, with shaded regions denoting the standard deviation.

## C.2 Online Learning

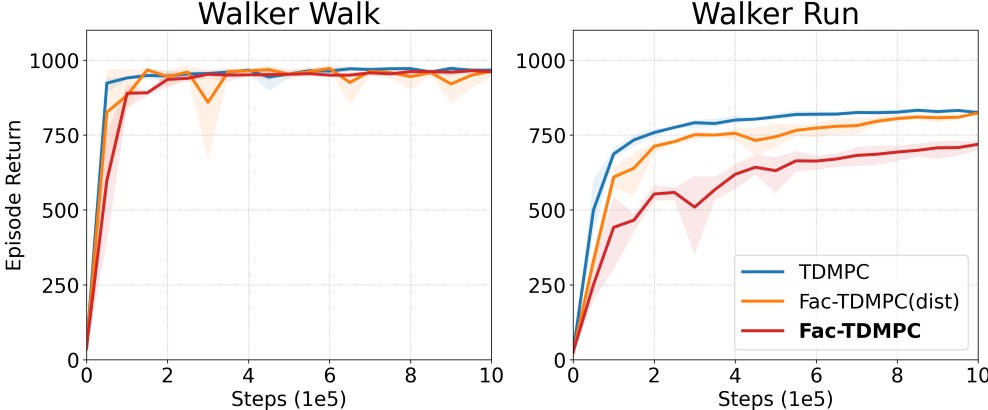

Figure 12: Online learning on two tas ks (`Walker Walk` and `Walker Run`). Each curve shows the average episode return over three seeds, with shaded regions denoting the standard deviation.

In this section, we evaluate Fac-TDMPC in an online learning setting, alongside its distillation-enhanced variant, Fac-TDMPC(dist) (see Section C.1), across two tasks of varying complexity. While the factored model is computationally more efficient, results demonstrate that for simpler tasks such as `Walker Walk`, all methods achieve comparable peak returns. However, on the more challenging `Walker Run` task, the purely online-trained Fac-TDMPC struggles to reach the optimal policy. Crucially, employing online model distillation enables the model to recover this performance gap, nearly matching the baseline's return. These results imply that factored models gain computational efficiency from a sreduced function space, yet may suffer from limited exploration in online settings—a constraint that can be effectively mitigated by distilling knowledge from a full model.

## C.3 Detailed Prediction Time

Table 3: Prediction time of different models on different robots. All wall times are reported in milliseconds with average and standard deviation over 100 runs. T1 represents testing on a single-thread machine.

| Methods | | Reacher | Hopper | Fish | Cheetah | Walker | Quadruped | Humanoid |
|---|---|---|---|---|---|---|---|---|
| **TDMPC** | $f_T$ | $5.83 \pm 0.01$ | $5.86 \pm 0.01$ | $5.83 \pm 0.01$ | $5.63 \pm 0.01$ | $5.86 \pm 0.01$ | $5.96 \pm 0.01$ | $5.83 \pm 0.07$ |
| | $f_R$ | $5.32 \pm 0.01$ | $5.34 \pm 0.01$ | $5.26 \pm 0.01$ | $5.06 \pm 0.01$ | $5.28 \pm 0.01$ | $5.34 \pm 0.01$ | $5.25 \pm 0.01$ |
| | $f_Q$ | $11.73 \pm 0.02$ | $11.70 \pm 0.04$ | $11.56 \pm 0.04$ | $11.35 \pm 0.05$ | $11.71 \pm 0.04$ | $11.86 \pm 0.05$ | $11.71 \pm 0.06$ |
| **10% TDMPC** | $f_T$ | $0.32 \pm 0.01$ | $0.33 \pm 0.01$ | $0.33 \pm 0.01$ | $0.32 \pm 0.00$ | $0.32 \pm 0.01$ | $0.32 \pm 0.01$ | $0.32 \pm 0.01$ |
| | $f_R$ | $0.25 \pm 0.00$ | $0.25 \pm 0.01$ | $0.25 \pm 0.01$ | $0.25 \pm 0.00$ | $0.25 \pm 0.01$ | $0.25 \pm 0.01$ | $0.25 \pm 0.01$ |
| | $f_Q$ | $0.85 \pm 0.01$ | $0.85 \pm 0.02$ | $0.88 \pm 0.03$ | $0.88 \pm 0.03$ | $0.87 \pm 0.02$ | $0.87 \pm 0.01$ | $0.85 \pm 0.03$ |
| **TAP** | $f_T$ | $1.78 \pm 0.00$ | $1.78 \pm 0.00$ | $1.78 \pm 0.00$ | $1.78 \pm 0.00$ | $1.78 \pm 0.00$ | $1.78 \pm 0.00$ | $1.78 \pm 0.01$ |
| | $f_R$ | $1.50 \pm 0.01$ | $1.54 \pm 0.01$ | $1.56 \pm 0.01$ | $1.53 \pm 0.01$ | $1.50 \pm 0.01$ | $1.49 \pm 0.00$ | $1.56 \pm 0.00$ |
| | $f_Q$ | $3.68 \pm 0.02$ | $3.81 \pm 0.03$ | $3.80 \pm 0.02$ | $3.69 \pm 0.03$ | $3.69 \pm 0.02$ | $3.67 \pm 0.02$ | $3.69 \pm 0.03$ |
| **Fac-TDMPC** | $f_T$ | $1.25 \pm 0.01$ | $0.83 \pm 0.02$ | $0.77 \pm 0.01$ | $0.75 \pm 0.03$ | $0.72 \pm 0.01$ | $0.63 \pm 0.01$ | $0.42 \pm 0.01$ |
| | $f_R$ | $1.02 \pm 0.00$ | $0.65 \pm 0.01$ | $0.57 \pm 0.01$ | $0.49 \pm 0.03$ | $0.48 \pm 0.01$ | $0.49 \pm 0.00$ | $0.32 \pm 0.01$ |
| | $f_Q$ | $2.56 \pm 0.03$ | $1.71 \pm 0.03$ | $1.53 \pm 0.03$ | $1.34 \pm 0.05$ | $1.32 \pm 0.03$ | $0.96 \pm 0.04$ | $0.69 \pm 0.03$ |
| **Fac-TDMPC (T1)** | $f_T$ | $3.71 \pm 0.01$ | $3.18 \pm 0.01$ | $3.88 \pm 0.02$ | $4.50 \pm 0.02$ | $4.46 \pm 0.01$ | $4.44 \pm 0.01$ | $4.11 \pm 0.02$ |
| | $f_R$ | $2.29 \pm 0.02$ | $2.66 \pm 0.02$ | $2.82 \pm 0.03$ | $2.93 \pm 0.03$ | $2.82 \pm 0.03$ | $3.23 \pm 0.02$ | $3.29 \pm 0.01$ |
| | $f_Q$ | $7.73 \pm 0.03$ | $6.86 \pm 0.02$ | $7.72 \pm 0.02$ | $7.92 \pm 0.04$ | $8.12 \pm 0.03$ | $6.69 \pm 0.04$ | $6.94 \pm 0.05$ |

This is the detailed prediction time of all baselines as in Table 3.

## C.4  End-to-end Planning Time

We present the end-to-end planning time per step in Table 4, including the forward passes of the encoder, reward, value and transition functions within the MPPI iterations. For a fair comparison, the numbers of samples and iterations are set to 512 and 6 across all experiments. The testing hardware is set the same as in Section 5.3. Similarly to Table 3, Fac-TDMPC achieves a substantial reduction in per-step planning time compared to TDMPC, and the efficiency gain continues to improve with the increase in action dimensions. Its speed even matches 10% TDMPC for humanoid robots, but the control performance is much better as shown in Table 4.

Table 4: End-to-end prediction time per planning step of different models on different robots. All wall times are reported in milliseconds with average and standard deviation over 100 runs.

| Methods | Reacher | Hopper | Fish | Cheetah | Walker | Quadruped | Humanoid |
|---|---|---|---|---|---|---|---|
| **TDMPC** | $203.66 \pm 0.44$ | $203.71 \pm 0.59$ | $203.84 \pm 0.32$ | $204.13 \pm 0.30$ | $204.01 \pm 0.27$ | $204.02 \pm 0.27$ | $204.41 \pm 0.83$ |
| **10% TDMPC** | $14.59 \pm 0.07$ | $14.58 \pm 0.08$ | $14.62 \pm 0.07$ | $14.65 \pm 0.06$ | $14.65 \pm 0.05$ | $14.71 \pm 0.12$ | $14.84 \pm 0.07$ |
| **TAP** | $64.94 \pm 0.05$ | $64.97 \pm 0.07$ | $64.94 \pm 0.0$ | $64.98 \pm 0.0$ | $65.02 \pm 0.14$ | $65.06 \pm 0.07$ | $65.10 \pm 0.14$ |
| **Fac-TDMPC** | $43.86 \pm 0.09$ | $29.30 \pm 0.10$ | $26.02 \pm 0.08$ | $22.92 \pm 0.09$ | $21.90 \pm 0.06$ | $21.34 \pm 0.05$ | $14.02 \pm 0.07$ |
| **Fac-TDMPC (T1)** | $132.33 \pm 0.30$ | $114.48 \pm 0.35$ | $131.78 \pm 0.42$ | $137.57 \pm 0.55$ | $129.70 \pm 0.48$ | $149.02 \pm 0.32$ | $142.93 \pm 0.77$ |

## C.5  Ablation Study

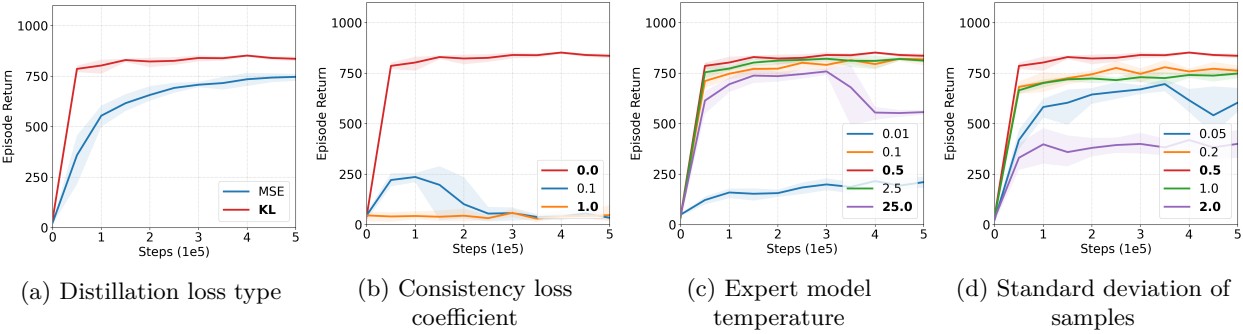

(a) Distillation loss type  (b) Consistency loss coefficient  (c) Expert model temperature  (d) Standard deviation of samples

Figure 13: Ablation study on key design choices of Fac-TDMPC on challenging (`Walker Run`, `Quadruped Run`, and `Humanoid Walk`) tasks. Each curve shows the average episode return over three seeds, with shaded regions denoting the standard deviation.

**Key Design Choices.** We conduct an ablation study on the three most challenging tasks (`Walker Run`, `Quadruped Run`, and `Humanoid Walk`) to assess the impact of key design choices. As shown in Figure 13, the KL-based loss consistently outperforms MSE, highlighting that ranking the optimal action is more efficient than regressing the exact value in policy distillation (Rusu et al., 2016). Within the KL setup, moderate values of both the noise standard deviation and temperature play crucial roles: a larger temperature smooths the target, while a smaller temperature overly emphasizes one particular action; a larger standard deviation introduces too much noise to the learning, while a smaller standard deviation causes overfitting of the optimal actions in the dataset. Surprisingly, adding a consistency loss as in TDMPC (Hansen et al., 2022) actually harms performance. As mentioned in Section 4.2, Fac-TDMPC actually aims to learn an equivalent world model in the $Q$-irrelevance ($\phi_Q$) and reward prediction (RP) abstraction instead of the next latent state prediction (ZP) abstraction. It indicates that the temporal consistency of the latent states is overly restrictive for the factored structures and unnecessary to achieve good planning performance.

