# OpenReview forum: "Fac-TDMPC: A Factored World Model for Robot Planning"
_TMLR — Accepted by TMLR_

### Review · Reviewer_k55R · 2025-11-04

**Summary Of Contributions:**

The paper introduces Fac-TDMPC, a model-based reinforcement learning framework that factorizes the latent world model of TDMPC/TDMPC2 along action dimensions. Transitions, rewards, and values are learned per action (or joint) and summed, enabling decentralized, parallel planning via model predictive path integral control (MPPI) with complexity reduced from $O(|A|^H)$ to $O(\sum_i |A_i|^H)$. Knowledge is transferred from a centralized expert through a distillation objective that preserves action-ranking by minimizing the KL divergence between soft reward/value distributions under Gaussian action perturbations. Empirically, the method achieves large planning speedups while matching or exceeding control performance on DMControl and MuJoCo tasks, also showing improved robustness, interpretability, and multi-task transfer.

**Audience:**

Yes

**Audience Explanation:**

The paper is well aligned with the interests of the TMLR audience. Planning latency is a major bottleneck for real-time robotics when using latent world models. The paper presents a pragmatic and elegant solution that combines structured factorization with ranking-preserving distillation to maintain accuracy while significantly reducing planning time. This balance between computational efficiency and control performance is the central contribution of the work. The contribution is likely to attract readers working on scalable MBRL, robotic control, and modular world models.

**Broader Impact Concerns:**

The proposed framework advances real-time model-based control by significantly reducing computational latency, which can improve safety and efficiency in robotic systems. The method also promotes modularity and interpretability, which are positive directions for trustworthy deployment. Potential risks include unsafe behavior under actuator or communication failures.

**Claims And Evidence:**

Yes

**Claims Explanation:**

The paper is clearly motivated and conceptually sound. It introduces an elegant “individual global max” proposition that formalizes when decentralized planning can be optimal under additivity and independence. The use of model distillation between centralized and factored models is pragmatic and theoretically well-founded. Empirically, the results are strong across seven robots and eleven tasks, showing substantial speedups without loss of control quality. The visualization of joint-level latent structures enhances interpretability, and the robustness and multi-task transfer results further validate the approach.

Moreover, the claims are well supported by accurate and convincing evidence, however mostly empirical. Planning speedups are demonstrated through direct wall-time measurements across robots, with results that scale consistently with action dimension. Control performance is validated by training curves on eleven tasks, showing that Fac-TDMPC matches or exceeds the centralized expert. Robustness to reduced MPPI sample and iteration budgets demonstrates smaller performance degradation compared to baselines. Ablation studies on temperature, noise, and loss type highlight the importance of the KL-based ranking objective and the decision to omit temporal consistency. Although the theoretical justification strictly holds under perfect independence, empirical evidence strongly supports the method’s effectiveness and scalability in practice.

**Requested Changes:**

The paper is solid overall and could require a few improvements to increase clarity and completeness. (1) Include an additional experiment or brief analysis illustrating how the method behaves when the transition independence assumption is mildly violated, to assess robustness under coupled dynamics. (2) Add a concise discussion or theoretical note explaining how deviations from full factorization may affect optimality or performance, maybe by discussing the very finite sample guarantees discussed in the motivation of the MBRL?. (3) Report end-to-end wall-clock planning times, including encoder and MPPI steps, for several computational budgets, to better quantify the practical speedup. (4) Compare Fac-TDMPC to one or two structured or modular world model baselines, such as attention-based latent dynamics, even if only at a conceptual level.

---

### Review · Reviewer_qvwx · 2026-01-20

**Summary Of Contributions:**

This paper proposes Fac-TDMPC, a model-based reinforcement learning algorithm that learns a factorised and latent world model. That is, Fac-TDMPC learns separate transitions function, reward functions and value functions per individual action instead of joint functions across actions. This factorisation has the benefit that the framework of decentralised Markov decision processes (Dec-MDPs) can be applied for planning. Specifically, decentralised MDPs facilitate parallel planning across actions as long as the transitions are independent and the reward functions are decomposable. This property is exploited by Fac-TDMPC to significantly speed up planning time compared to the joint world models used by vanilla TDMPC. Importantly, Fac-TDMPC learns its world model through model distillation, usually from an expert (joint) model obtained from TDMPC. The experiments illustrate significant improvements in prediction time across a range of standard robotics benchmarks while maintaining comparative or improved performance with respect to more expensive joint world models. Fac-TDMPC additionally provides seemingly more robust control performance when applying certain perturbations of the actions.

**Additional Comments:**

Thank you for also discussing limitation of your method by showing that there are cases where factorisation of the world model can avoid learning certain complex situations, e.g. Figure 9. It will be interesting to see how this limitation manifests itself in more complex benchmarks.

**Audience:**

Yes

**Audience Explanation:**

The findings of this paper are surely of interest to the model-based RL community as it showcases significantly improved results in terms of planning efficiency. They also show that a reduction of theoretical expressivity through action factorisation of the world model can still maintain high performance, which would be the first expected disadvantage of the loss of expressivity. In this sense, the results are surprising and interesting.

**Broader Impact Concerns:**

No concerns on ethical implications.

**Claims And Evidence:**

Yes

**Claims Explanation:**

The main claim of increased computational efficiency while maintaining control performance is substantiated by the experimental discussion in sections 5. Specifically, Figure 4 shows that the performance in terms of average episodic return is comparable or better to TDMPC, the fully expressive, yet expensive baseline. Simultaneously, Figure 6 shows significantly improved predictions times (the time for one model call) as well. This prediction time *seems to be* the main contributor to overall computation time, because both TDMPC and Fac-TDMPC perform well for the same number of model calls on a selected number of experiments. Potentially, Fac-TDMPC might even work just as well with less model calls.

The remaining secondary claims of robustness to action perturbations and explainability are supported by the discussions in sections 5.5 (robustness) and 5.6 (explainability). However, it is unclear if the type of perturbations discussed in section 5.5 are of interest to the community. Additionally, it is also unclear whether the type of explanations through t-SNE visualisations and H-step return plots per action are of interest. Both do provide some evidence of robustness and explainability though.

**Requested Changes:**

I do have a number of questions and remarks. The following are important to recommending acceptance.

1. My main question is why you limit Fac-TDMPC to the special case of learning to distil its world model from a given expert model. Does the methodology not apply to directly learn, in an online fashion, from trajectories gathered during exploration? That is, is Fac-TDMPC not applicable to the same problem as the vanilla TDMPC? If not, why? If yes, why do you only consider learning from an expert model? Do you expect the factorisation would inhibit learning usable world models? This question is especially pertinent considering that you do consider an online distillation task in the appendix.
2. Please extend Figure 6 with a version of Fac-TDMPC running on a single thread as well. This comparison is useful to consider the scaling behaviour under limited computational resources. This is particularly important for harder tasks with more actions where the number of actions exceeds the available threads. It does not undermine the computational complexity advantage as it should rather illustrates the advantage of going from $\mathcal{O}(|\mathcal{A}|^H)$ to $\mathcal{O}(\sum_{i = 1}^N |\mathcal{A}_i|^H)$.
3. Are there references for the type of action perturbations that you consider? Similarly, do you have references to indicate that the type of T-SNE visualisations or returns graph analyses actually contribute to explainability in MBRL? Are these of interest to the community?
4. The multi-task learning discussion is severely lacking in exposition. To promote a self-contained paper, certain experimental details should be specified. For one, how specifically does Fac-TDMPC apply to the multi-task setting? How does its world model look like? Since you factorise it per action, do you construct a specific world model per action per task? Do you make use of a task embedding as TDMPC2? If so, how is that integrated. Right now, it is impossible for the reader to see what the results in section 5.7 imply, which makes the section useless. A bit more information about the multi-task dataset in the appendix would also help.

The following comments would simply strengthen the work.

1. Could you elaborate on why Fac-TDMPC gains computational efficiency for increased action dimensions? Is this purely because of exploiting more parallelism? There should be a plateau here then as well once you run out of threads, which deserves a couple of words.
2. More of a confirmation question; you don't enforce the ZP property when learning TDMPC, right? Otherwise this might skew the discussion in Figure 7 towards Fac-TDMPC due to Fac-TDMPC being allowed to be temporally disconnected from the original world model.
3. Where are the standard deviations of Figures 6, 7, 8 and 10?
4. Typo "dyanmics" in the legend of Figure 2.

---

### Review · Reviewer_1HW5 · 2026-04-03

**Summary Of Contributions:**

Fac-TDMPC proposes factoring the latent-space model in TDMPC into per-action-dimension sub models, including transition, reward, and value, which is framed as a factored decentralized MDP. The factored structure is learned via KL-based model distillation from a trained centralized TDMPC expert. The key payoff is decentralized MPPI planning yielding substantial speedups with minimal performance loss across DMControl tasks.

Strength:

1. The core idea is clean and well-motivated. Treating each action dimension as an independent "agent" in an fDec-MDP and enforcing additive reward/value decomposition is a natural way to break the curse of dimensionality.

2. The distillation procedure is a sensible design choice.

3. The multi-task results show good results on the speed ups over the baselines.

Weakness:

1. The independence assumption is extremely strong, and the paper doesn't stress-test it enough. The entire framework rests on transition independence and additive reward decomposition across action dimensions. For the manipulator example this holds by construction, but for locomotion tasks it's clearly violated in the ground-truth dynamics, i.e.,  hip torque and knee torque on the same leg produce highly coupled effects on the body's CoM. The paper acknowledges this in text (Section 4.2: "we impose monotonicity... across all states") but doesn't really elaborate or discuss with when and why the factored model works despite the assumption being wrong.

2. The paper evaluates Fac-TDMPC exclusively in a distillation setting: a centralized TDMPC expert is fully trained first, then the factored model is distilled from its replay buffer. This means the results can't distinguish between: (a) the factored structure is a good inductive bias for dynamics modeling, and (b) the factored structure is a good compression target given a pre-trained centralized model. These are very different claims, and the paper's framing leans toward (a) while the evidence only supports (b).

3. The benchmark suite is limited to DMControl, and the tasks are relatively low-dimensional. Furthermore, all tasks use proprioceptive observations, no image-based tasks are included, despite TDMPC supporting them.

**Audience:**

Yes

**Audience Explanation:**

I think the topic is interesting, extracting latent structure of the dynamics, albeit the methodology is not very principled.

**Broader Impact Concerns:**

None.

**Claims And Evidence:**

Yes

**Claims Explanation:**

There are some misalignment between the framing of the problem and the solution methods this work provides. See weakness 2.

**Requested Changes:**

I hope the authors can address the scope concern, particular w.r.t to TDMPC, which is used as an expert to distill the model learning. This paper presents to me first as a method to learn the structure on its own without any teacher policy but it turns out it needs a precursor, which is disappointing during reading.

---

### Decision · Action_Editor_yN6s · 2026-05-25

**Recommendation:** Accept as is

**Audience:**

Yes

**Audience Explanation:**

Yes, the planning / RL sub community of TMLR would be interested in this.

**Claims And Evidence:**

Yes

**Claims Explanation:**

The paper proposes Fac-TDMPC, a factored latent world model for model-based RL that decomposes transition, reward, and value prediction across action dimensions or joints. This enables decentralized MPPI planning, substantially reducing planning latency compared with centralized TDMPC-style world models, while largely preserving control performance. Overall, the discussion converged toward acceptance. The main concerns were: (a) that real robot dynamics, especially locomotion, violate action independence; (b) that the original framing implied Fac-TDMPC might learn useful structure from scratch, but the evidence mainly supported distilling from a centralized expert; (c) limited benchmarks, with reliance on DMControl/MuJoCo-style proprioceptive tasks, with no image-based experiments.

The authors responded by reframing the paper more clearly as an efficient planning/compression method. They added experiments showing that online distillation can track the centralized expert during training, while pure online Fac-TDMPC works on simpler tasks but struggles on harder ones. They also added single-thread results, end-to-end timing, a discussion of failure modes, and standard deviations for plots.